# Fast detection of slender bodies in high density microscopy data

Albert Alonso [1] & Julius B. Kirkegaard [1✉]

Computer-aided analysis of biological microscopy data has seen a massive improvement with the utilization of general-purpose deep learning techniques. Yet, in microscopy studies of multi-organism systems, the problem of collision and overlap remains challenging. This is particularly true for systems composed of slender bodies such as swimming nematodes, swimming spermatozoa, or the beating of eukaryotic or prokaryotic flagella. Here, we develop a end-to-end deep learning approach to extract precise shape trajectories of generally motile and overlapping slender bodies. Our method works in low resolution settings where feature keypoints are hard to define and detect. Detection is fast and we demonstrate the ability to track thousands of overlapping organisms simultaneously. While our approach is agnostic to area of application, we present it in the setting of and exemplify its usability on dense experiments of swimming *Caenorhabditis elegans*. The model training is achieved purely on synthetic data, utilizing a physics-based model for nematode motility, and we demonstrate the model's ability to generalize from simulations to experimental videos.

[1] Niels Bohr Institute & Department of Computer Science, University of Copenhagen, Copenhagen, Denmark. ✉email: julius.kirkegaard@nbi.ku.dk

Large-scale, high-throughput quantification of microscopy data has increasingly become possible with the aid of computer vision[1–6]. In particular within the last decade, deep learning techniques[7–9] have improved and enabled accurate image analysis of microscopy data in a broad range of areas including cell counting[10,11], cell segmentation[12–14], nucleus detection[6,15], sub-cellular segmentation[16], drug discovery[17], cancer detection[18–20], and the identification of infectious diseases[21,22]. Detection models serve as the fundamental operation in tracking procedures, and combined with suitable tracking algorithms, these can achieve morphologically resolved organism tracks that can accurately quantify organism motility[23], the application of which ranges from fundamental neuroscience[24–26] and the circuitry of simple organisms[27–30] to drug discovery[31–35].

Multi-organism detection can be achieved at increasing levels of fidelity: at the crudest, only center-of-mass locations or bounding boxes are predicted[36] which does enable tracking of organisms but provide little morphological information. In contrast, pixel-wise segmentation models[12] and pose estimation using keypoints[37] reveal accurate shape dynamics when employed on high-resolution data. However, these methods rely on high definition objects, as segmentation and prediction are highly sensitive to noise. In particular for organisms that are long and slender, pixel-wise segmentation fails at low resolution as correct predictions require sub-pixel accuracy. Moreover, at high densities, these methods may fail due to their inability to properly handle overlap between organisms.

Here, we consider the problem of studying slender organisms at low resolution and high density with the goal to enable both accurate identity tracking and quantification of shape dynamics. This problem has traditionally been approached by employing pixel-wise segmentation and subsequent skeletonization procedures[38–43], an approach that requires model-based approaches[44,45] or ad-hoc procedures[46] to solve the problem of correctly identifying overlapping organisms, the combinatorial complexity of which blows up at high densities. To this end we abandon pixel-wise output and instead construct a neural network architecture that predicts, potentially overlapping, centerlines directly[47–49]. Our method enables both accurate shape prediction and tracking in dense experiments of slender objects, a key challenge for a broad class of systems (Fig. 1), including tracking of nematode worms[50–52], spiral or elongated bacteria[53–56], spermatozoa[57,58], the flagella of both eukaryotes[42,43] and prokaryotes[59], and freely swimming flagella such those of microgametes[60].

Our method relies on recent advances in deep learning[61–65] and extends these by a few simple ideas: In still micrographs, the identities of individual worms can end up being obscured by overlaps making them impossible to accurately identify, and only by relying on the adjacent frames can they be correctly resolved. Thus, to allow the neural network to encode the identity of individual bodies as a function of their motion, the input to our neural network is taken to be short video clips rather than single frames. Our network outputs multiple independent predictions, and for each produces (1) the centerline of the organism, (2) an estimated confidence score for the prediction, and (3) a latent vector, the space of which we induce a metric on that measures whether two predictions are trying to predict the same body. To train the network, each output quantity is associated with a specific loss term, where, importantly, the centerline loss term is permutation-invariant in the labels. To resolve overlap, we do non-max suppression[36], but rather than measuring distances between curve predictions, we use the latent space output, which allows two predictions to be kept even though they are close in physical space. This enables correct predictions for data in which objects overlap very closely. Our method is further tailored to

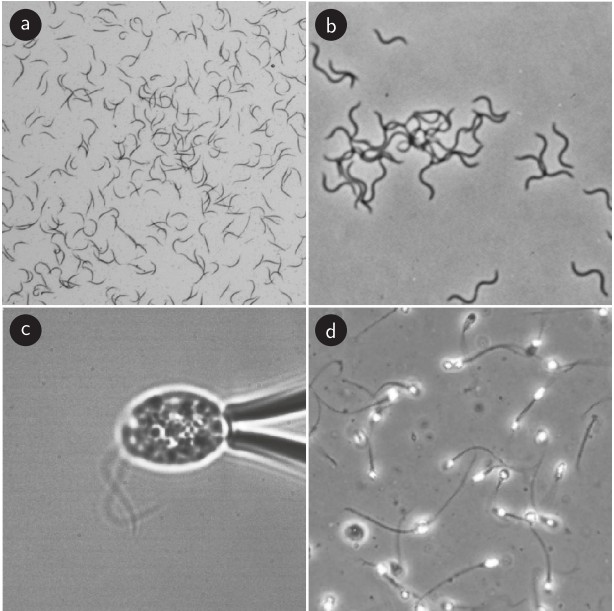

**Fig. 1 Microscopy images of different microorganisms whose slender structure and frequent overlaps makes them hard to detect using classical approaches. a** *C. elegans* motility experiment from the dataset of this paper. **b** Motile, flexuous, thin, spiral-shaped *B. pilosicoli* bacteria. Still from ref. [55], with permission. **c** Beating flagella of the green alga *C. reinhardtii*, provided by Kirsty Wan, University of Exeter. **d** Swimming human *spermatozoa*. From dataset in ref. [58].

support the subsequent tracking process, which must link uniquely predictions from frame to frame. To that end, we not only predict the object location at a single timepoint, but also predict consecutive past and future centerlines. Using these time-resolved predictions in the linking process enables high-precision tracking even through dense regions.

Our method is in-principle applicable to all microscopy datasets that involve slender bodies, but we do not develop its general applicability here. Instead, we focus on its applications for tracking dense experiments of swimming *C. elegans* worms, a popular model system in neuroscience[66], human diseases[67], drug discovery[32], motor control[68], memory[69], and ageing[70]. Studies of *C. elegans* often rely on phenotypic assays that measure the motility of the nematode worms as a function of some environmental condition or treatment[35,71–83], the throughput of which can be massively increased if overlap between organisms can be tolerated. Likewise, resolving identities of organisms during overlap is crucial for studies of interactions between organisms[84]. Previous work on tracking *C. elegans* have generally employed classical computer vision approaches to accurately track single or a few high-definition worms[39,85–89], or many low-resolution worms at non-overlapping densities[40,90,91], in some cases by utilizing a computational model of the worm motion for hypothesis tracking[39,44,45,85].

Recently, deep learning techniques have been utilized to track *C. elegans* worms using e.g. bounding box predictions[92–94] and fully resolved centreline in the case of isolated worms[95], allowing for detection also during periods of self-overlap.

With this paper, we publish a dataset of videos of motile *C. elegans* worms imaged at a wide range of densities. The dataset includes~ 1,500 labeled midlines that we use to evaluate, but not train, our detection model. We demonstrate that our model can be trained exclusively using synthetically generated data and yet generalizes well to real videos. Our method leverages the parallel capabilities of convolutional neural networks and is thus able to

handle thousands of detections in a single pass, resulting in real-time detection at ~90 Hz at 512 × 512 resolution on a single GPU. The code is open source and available at https://github.com/kirkegaardlab/deeptangle.

## Results

**Architecture**. Our model is based on single-stage detection models[36,61] that output many candidate predictions per target in a single forward pass and rely on a score system to prune until a single candidate is left for each target object. The performance of such single-stage models has been shown to enable accurate real-time bounding box detection[64]. Figure 2 illustrates the overall structure of our approach. The backbone of our neural network (Fig. 2a) consists of convolutional residual networks[62] and the output of our model is composed of a set of centerline predictions $z = [x^-, x, x^+]$ representing the past, present, and future motion of the bodies. We represent the centerlines by $k$ equidistant points along the center of the body [Fig. 2d]. The centerlines contained within the set maintain alignment with a consistent head positioning across the three predictions. In addition, the model outputs confidence scores $s$ and latent vectors $p$ that are used for subsequent filtering (Fig. 2b) (see Methods).

We take the input to our model to be a stack of consecutive frames in order to provide the model with a temporal context (Fig. 2c). In the present case of motile slender objects where dynamic crossings and overlap between objects are very common,

a temporal context can provide the necessary information to resolve the problem of correct identification. Furthermore, the temporal context allows the output of our model to include information on the motion of the centerlines, which we will further exploit for tracking purposes.

The backbone of our neural network performs a $16^2$-fold reduction in resolution when mapping the input images to feature space, from which the network outputs multiple anchored predictions. This anchored approach means that the only restriction on input size is that its dimensions be divisible by 16, and, in particular, it allows training at a certain resolution $H \times W$ and subsequent inference at another $H' \times W'$ without loss of accuracy. We choose the resulting number of candidates to be considerably larger than the number of objects in the frame, thus ensuring that all objects have suggestions.

**Detection on dense *C. elegans* experiments**. To evaluate our approach, we study microscopy videos of swimming *C. elegans* worms. We are particularly interested in videos captured at much higher densities than those typically used in motility experiments. Thus we evaluate our model on wide-field videos captured under approximately uniform illumination[40], exemplified in Fig. 3a. In our dataset, the number of nematode worms varies ranging from ~400 with a small probability of overlap occurring (≈0.05 average overlaps per worm) to extremely densely packed plates with up to ~6000 nematodes, where there is, on average, one overlap per

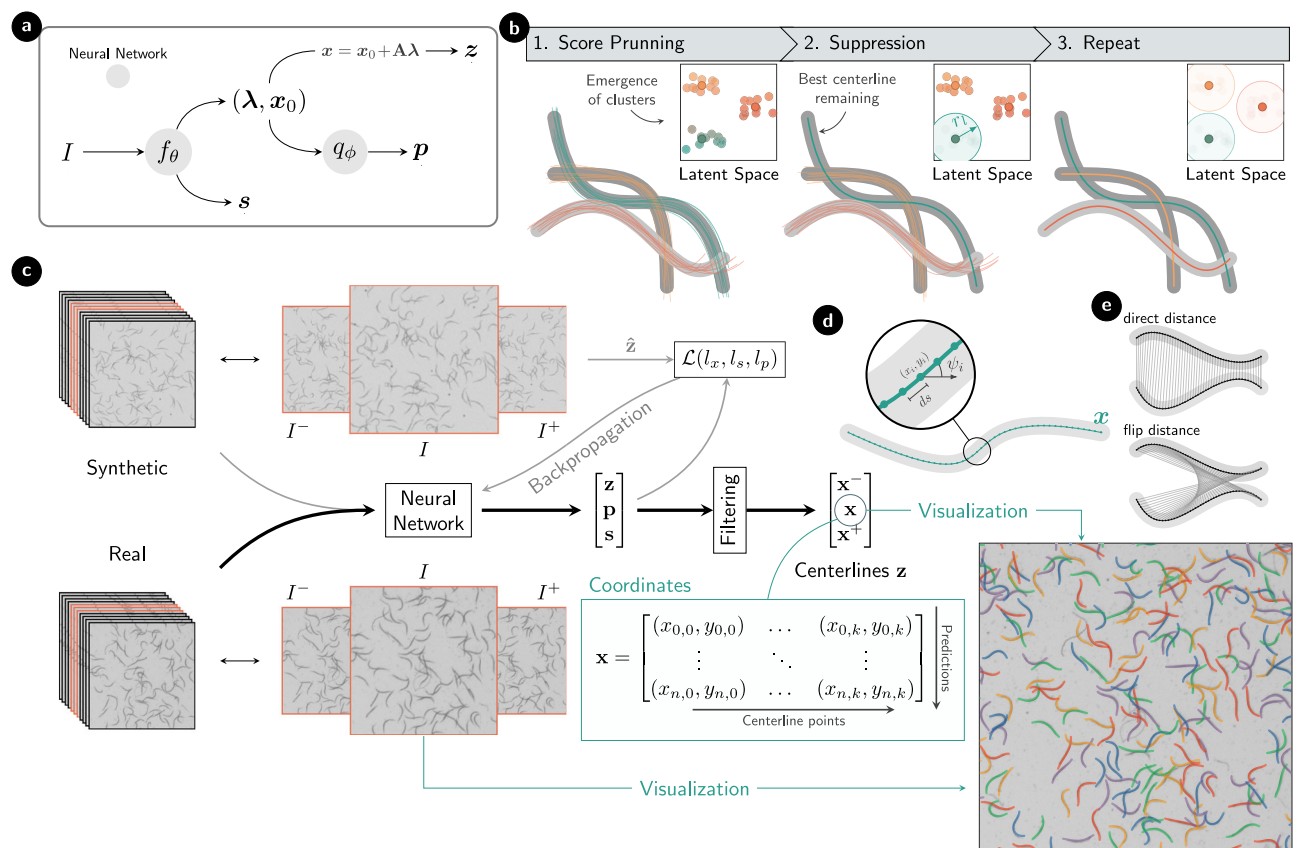

**Fig. 2 Method workflow. a** Structure of the detection method. Trainable neural networks are colored in gray, and represent the convolutional neural network $f(I; \theta)$ and the latent space encoder $q(\lambda, x_0; \phi)$. **b** Procedure to prune unfiltered predictions to final detections with the use of the encoded latent space vectors. **c** Method overview from the input clip $I$ (we use a stack of 11 frames in this work) to the final matrix of centerlines $x$. The target frames $[I^-, I, I^+]$ (center frames from the clip, orange) are explicitly shown for both the synthetic and real videos. In addition, the training setup is represented using lighter color arrows; from synthetic data to loss backpropagation. After detection, direct visualization of the predicted centerlines $x$ is possible. **d** Diagram with a centerline descriptor composed of $k$ equidistant points along the skeleton of the nematode. **e** Visual representation of the two distances used in Eq. (2), the minimum of which corresponds to correct head-tail alignment and is the one that will be used in the model.

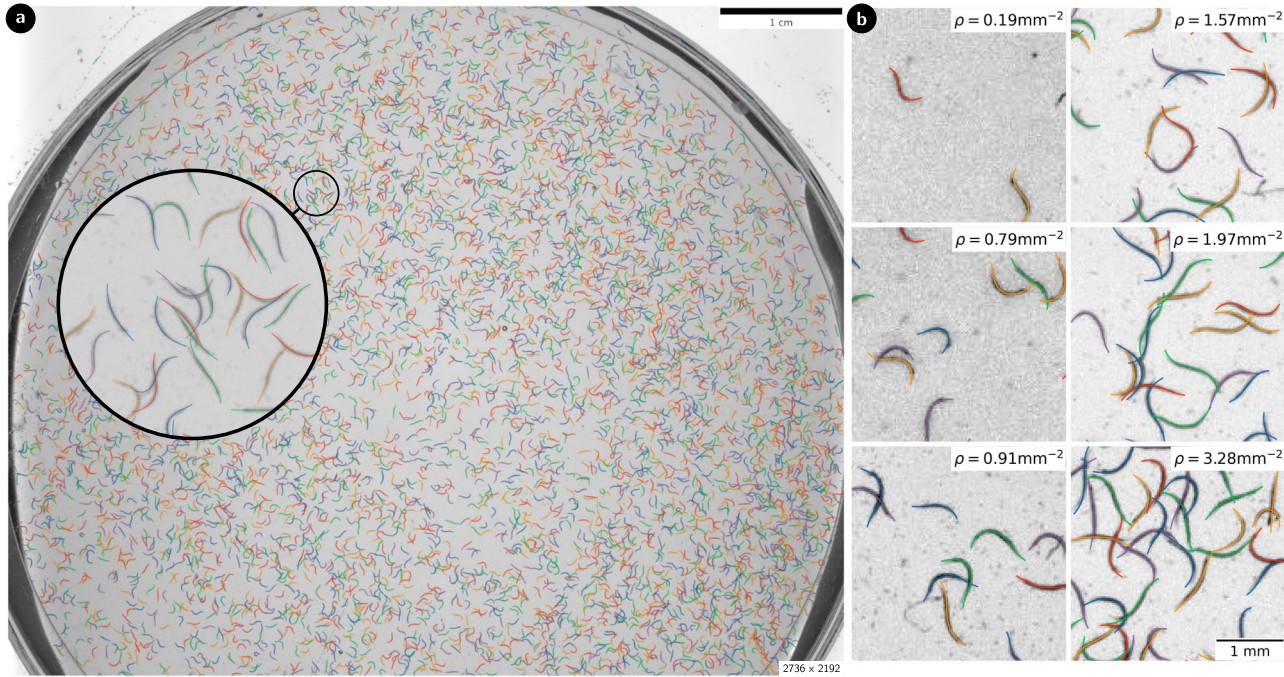

**Fig. 3 Qualitative showcase of the capabilities of the model. a** Detected centerlines predicted on an entire densely populated well plate with a single forward pass through the neural network. Inset shows a zoom-in section to demonstrate the accuracy of detection across the entire plate (except near borders, where the plate interferes). The total plate contains around 6000 detections. **b** Close-up evaluation of different experimental clips with different densities of worms.

worm. This means that in the dense plates, detection methods that stop tracking after contact between worms happens are rendered completely ineffective.

Defining worm density $\rho$ as the number of worms in a region per square millimeter, we find, as expected, a linear relation between the average amount of overlap per worm and the density (Fig. 4a). Due to the spatial heterogeneity of the worm distribution inside the plate, higher densities can be observed when considering small regions. On 100 mm$^2$ scales, the highest density in the dataset is $\rho \sim 2.5$ mm$^{-2}$, but this jumps to an extreme $\rho \sim 3.5$ mm$^{-1}$ when considering 10 mm$^2$ regions, where humans begin to struggle to correctly identify worms. For quantitative evaluation of our model, ~200 random regions of the videos were sampled and hand-labeled resulting in ~1500 labeled worm centerlines. A sample of frames is shown in Fig. 3b to provide a sense of the different densities encountered in the evaluation dataset, with the predictions of the model overlaid.

To train our network, we implement a physics-based synthetic dataset generator to exploit perfectly defined labels (see Methods). This approach removes the need for a supervised dataset, and also allows labeled videos in situations where manual labeling may not be reliable, or where the subjectivity of the human labellers can result in inconsistent labels. Physics-based synthetic datasets have successfully been used to train systems on similar conditions, for instance where manual labeling may introduce unnecessary noise or bias to the model[16]. Naturally, this requires the formulation of a physical model that is accurate on the relevant time scales. Furthermore, dependence on synthetic datasets can result in a divergence between the target and the training data, potentially leading to inaccurate predictions during inference—a phenomenon that is avoided if the model is trained on real data.

*Performance.* Despite being trained exclusively on synthetic data, the model's inference performance is very good on real clips. From visual inspection, no immediate discrepancies are observed between detections in low density clips and at high density

(Fig. 3b). Likewise, per design, the network accuracy is independent on the input clip dimensions, and the parallel structure of convolutions permits the use of large videos covering thousands of nematodes to be processed simultaneously in a single forward pass (Fig. 3a). We note, however, that even though no quality impact on detections is observed when using large fields-of-view clips, there can be a dependency if non-uniform illumination is used as different sections of the frame may have different requirements for preprocessing.

For a quantitative assessment of the method accuracy, we compare to the manually labeled dataset, an example of which alongside the model predictions can be seen in Fig. 4c. As the predictions are densely defined centerlines (here, ~50 points), we used an asymmetric version of dynamic time warping $\delta_{\mathrm{adtw}}$ (defined in "Methods" and illustrated in Fig. 4b) to evaluate the accuracy of the predictions using labels with lower fidelity.

The results of evaluating the trained model on the labeled dataset are shown in Fig. 4. For reliable comparisons, we first solve the assignment algorithm for the label-prediction pairs. This means that in the case of two completely overlapped worms, two predictions need to be present to not count as a miss, and likewise, two predictions cannot be considered to target the same label. We find an average error of $\delta_{\mathrm{adtw}} \approx 0.54$ px with no strong dependency between accuracy and density of worms [Fig. 4d], with the exception of a slight increase in error for extremely dense clips (~3.5 mm$^{-2}$). The average error corresponds to less than the width of a worm ($\approx 2$ px $\approx 50$ μm), and part of this can be attributed to the fact that human accuracy is also near the half-pixel level (Fig. 4c). Some outliers can be seen however, which can mostly be attributed to an artefact of the model, where the network mistakes a single long worm for two overlapping shorter predictions. This effect seems particularly sensitive to incorrect intensity normalization of the videos.

Let $\sigma_\epsilon$ be a cutoff distance above which we no longer consider the predictions to be targeting the closest label. For all the figures in Fig. 4, this cutoff is assumed to be $\sigma_\epsilon = 3.0$ px, and we observe

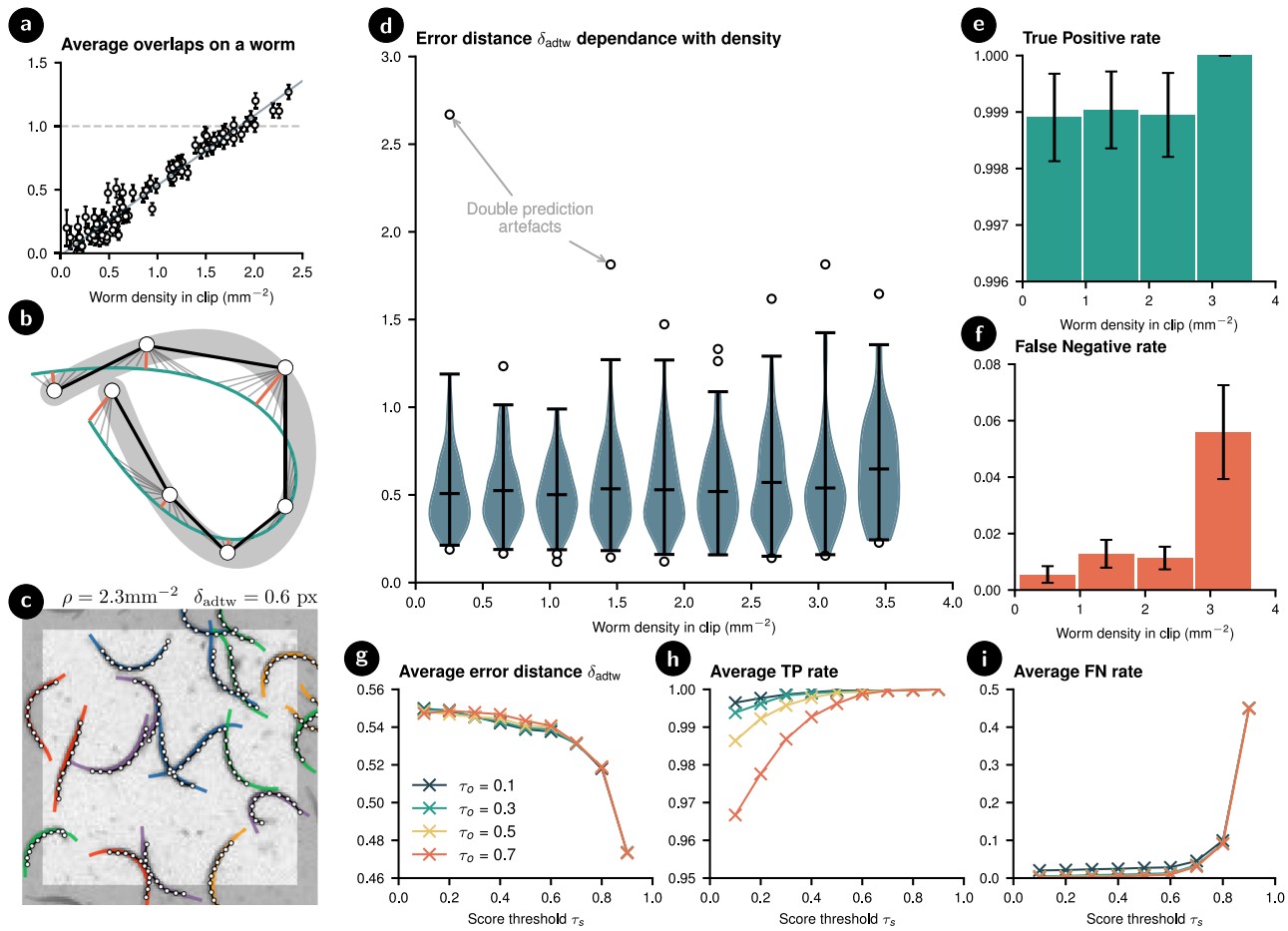

**Fig. 4 Quantitative performance metrics on the detection of slender bodies in dense experiments of swimming nematodes. a** Average number of overlaps counted on frames of pixel size $512 \times 512$ with different densities of worms ($N = 90$). **b** Illustration of the asymmetric dynamic time warping distance error corresponding to the average value of the orange euclidean distances between the prediction (green) and the labeled points (white). **c** Example frame with manually labeled points (white) and models predictions (colored). The metric is only evaluated in the lighter area of size $100 \times 100$. **d** Quantified accuracy of the detections by showing the distance to the manually labeled centerlines. Distributions for different densities are shown. The violin plots represent the 99 percentile of the data whereas outliers are plotted individually. **e** Rates for True Positive on the manually annotated dataset [$N = 364, 467, 441, 143$]. **f** Rates for False Negative on the manually annotated dataset [$N = 1825, 2078, 1902, 597$]. **g-i** Performance of the model with different combinations of score ($\tau_s$) and overlap ($\tau_o$) thresholds. $N = 1420$. Error bars indicate standard error.

no significant changes by tuning it within the range of sensible values. We define the True Positive (TP) rate as the fraction of predictions that both get assigned a label and this label is within the distance $\sigma_\epsilon$ of the prediction. Figure 4e shows that the model rarely predicts a centerline where there is nothing with a TP rate of 0.999. Nevertheless, there are some predictions that do not get assigned a label which can be attributed to the double-prediction artefacts just mentioned. The likelihood of this happening decreases with density, but the rate is so low that it is almost negligible. Similarly, we define the False Negative (FN) rate as the fraction of labels that are not assigned a prediction closer than $\sigma_\epsilon$. Figure 4f shows that the model in general manages a low FN rate at around ~0.015, but that this increases to a rate of ~0.06 at extreme densities such as $\rho \geq 3.0 \, \mathrm{mm}^{-2}$, where clusters tend to be densely packed and manual labeling likewise becomes challenging.

The filtering process depends on two user-defined thresholds: the score threshold $\tau_s \in [0, 1]$ is used to prune predictions with low confidence scores (Fig. 2b(1)) and the overlap threshold $\tau_o \in [0, 1]$ is used for filtering by setting the maximum probability of two independent predictions to be targeting the same object (Fig. 2b(2)). Throughout this paper, we have set these to $\tau_s = \tau_o = 0.5$. We evaluate how different combinations of thresholds may alter the

performance results. Figure 4g–i shows the average performance obtained across all densities when filtering the predictions with variable thresholds. In spite of some dependency between worm density and TP/FN rates, we consider the average metric to be a good indicator of the performance for each case.

Figure 4g shows the effect of the thresholds on accuracy. No significant dependency on the thresholds is observed. This can be explained by the fact that accuracy is determined by the best predictors only (through assignment), which are not discarded until a high $\tau_s$ is used, and once those are removed, $\tau_o$ becomes irrelevant. Further, the fact that there is no notable difference between different values of $\tau_o$ indicates that the clusters are highly compact.

In contrast, Fig. 4h shows that the TP rate has a stronger dependency on $\tau_o$ at low $\tau_s$ because low score predictions do not form compact clusters, and therefore a larger exclusion radius is required to discard them. Finally, Fig. 4i shows that misses only begin to occur once the best predictions are discarded, and a strong dependence on the $\tau_s$ is not observed before that point.

**Tracking from consecutive detections.** Motility assays require not only accurate detections but also the ability to link these across frames to form time-resolved tracks of individual organisms. This is challenging at high densities where we have the breakdown of

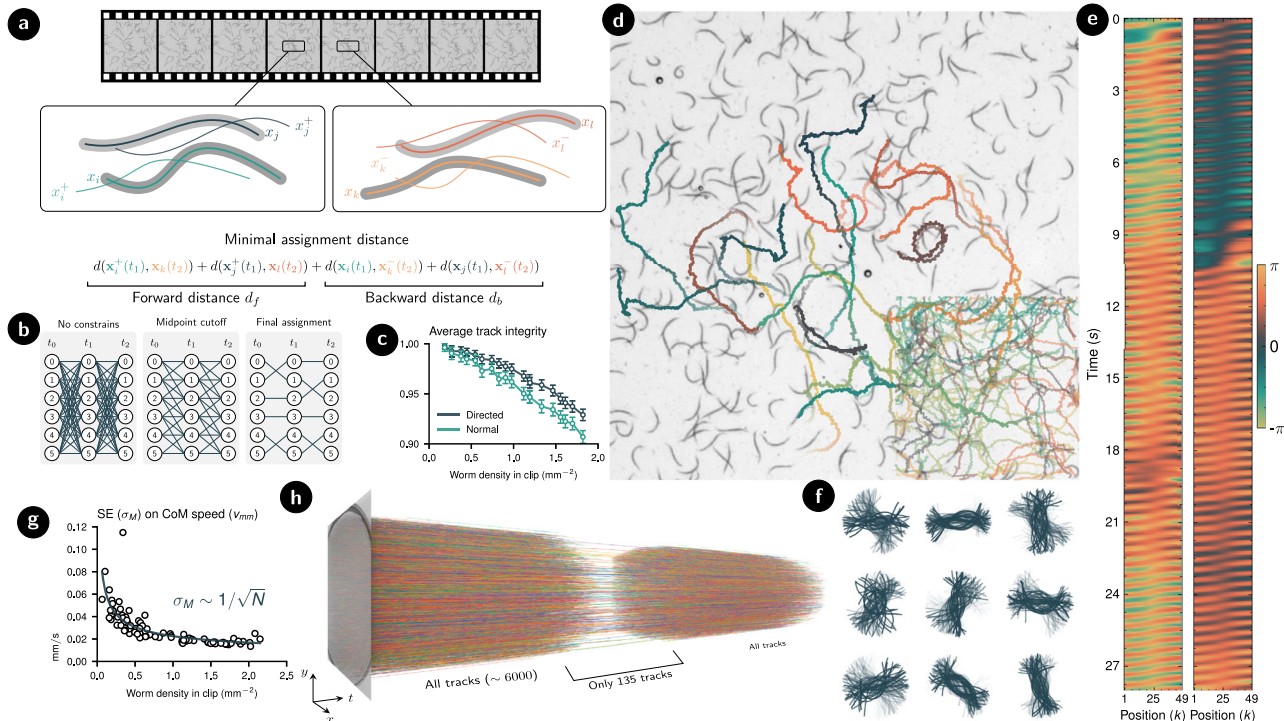

**Fig. 5 Tracking methodology and results. a** Illustration of the directed distance used to assign consecutive detections of the same identity. The simplified drawing shows two independent predictions at adjacent frames and showcases how the assignment scheme computes the identity by comparing future-present and past-present distances and choosing the assignment that minimizes their sum. **b** Diagram showcasing how using a location cutoff simplifies the assignment problem. Nodes represent independent detections at each frame whereas edge values are given by the directed distance measure. The assignment happens by minimizing the sum of edges at each timestep. **c** Comparison of using the straightforward centerline distance and the proposed directed approach. The accuracy is evaluated by measuring the integrity of the tracks. In contrast to other metrics in this paper, this plot has been obtained using synthetic worms as long-term, accurate tracks are required to evaluate the tracking integrity (See 4 for details on Tracking integrity). Error bars indicate standard error ($N$ ~ 300 at lowest density to $N$ ~ 3000 at highest density). **d** Qualitative example of 30 s trajectories of the center of mass of the nematodes in a dense experiment. The still background image represents the last frame of the video. To improve the visualization, a small subset of the trajectories is shown. In contrast, a corner of the frame is used to display all the trajectories to showcase the density of simultaneous tracks. **e** Two samples of the centerline angle $\psi$ of two randomly sampled nematodes from (**d**). **f** Undulations corresponding to 30 s of the detections relative to the center of mass coordinate of nine randomly sampled nematodes from (**d**). **g** Standard error value of the measurements of the center of mass speed as a function of density. **h** Showcase of the possible throughput of the method, by simultaneously tracking more than 6000 tracks from a full dense plate. A small window on the tracks is shown to showcase their continuity.

the assumption that the closest detected object to the previous frame corresponds to the same identity. In general, greedy approaches to particle tracking such as assigning directly the closest particle in consecutive frames frequently leads to failed tracks. Instead, the process of tracking can be efficiently formulated as a set of linear assignment problems[96]. Naturally, here we can expand upon particle tracking by using a metric that measures distances not between center-of-mass of the worms, but between the full centerlines (Fig. 2e). This works well for most predictions but can fail for fast-moving worms or in dense clusters.

A separate approach to tracking is Kalman filtering. This would require separate detection of entry and exit events of worms, as well as a probabilistic model for worm motility, which would most likely have to be highly non-linear. Kalman filtering is viable for the tracking of few organisms, but for present large-scale systems we require a more efficient approach. As our model also outputs centerlines from adjacent frames (to embed temporal information into the latent vector, see Methods), we propose a directed metric that leverages both past $x^-$ and future $x^+$ centerlines predictions (Fig. 5a). Thus to find a mapping $\sigma$ from one frame to the next, we solve

$$\sigma = \arg\min_{\sigma}\left[\sum_i d(\boldsymbol{x}_i(t), \boldsymbol{x}_{\sigma_i}^-(t')) + d\left(\boldsymbol{x}_i^+(t), \boldsymbol{x}_{\sigma_i}(t')\right)\right]. \quad (1)$$

Identity assignment can be seen as a network flow global optimization where nodes represent detections and edges carry the cost of assignment. To avoid having to perform all possible combinations of assignments, we include a physical distance threshold on the midpoint of the central line. This threshold significantly simplifies the assignment scheme and improves the runtime of the filtering process (Fig. 5b). Notice that due to the flip-invariance of the distance metric, consecutive assignments of sets of centerlines are not necessarily aligned, hence a trivial alignment of the centerlines is carried out during post-processing before being analyzed. Likewise, head-tail alignment with the real worm is not granted and a post-processing step would be required to guarantee the alignment, e.g., by using temporal information such as the direction of the undulation wave (Fig. 5e).

To quantify the performance of these methods, we define the *tracking integrity* ι as a scalar that indicates how consistent the assignment of a label to a prediction is along the tracked video. Perfect tracks have ι = 1, whereas labels that get assigned two different identities for half of the duration of the video have ι = $\frac{1}{2}$, and so on (see 4 for a detailed definition). We evaluate this on synthetically generated videos of 10 s (200 frames) that have perfectly labeled tracks, the results of which are shown in Fig. 5c. On videos with densities up to 2.0 mm$^{-2}$, we achieve an average

integrity of $\iota \approx 0.97$. This is ~30% improvement of the error over using direct detection assignment defined in Eq. (2). We observe that the integrity is almost perfect at low densities, but drops to $\iota \approx 0.93$ at the highest densities.

When applied to high density videos of *C. elegans*, the tracking method is able to keep track of individual worms as they pass through clusters of other worms (Fig. 5d) (see Supplementary Videos 1–3). In contrast to pixel-level classification of worms, our approach outputs centerlines directly, and thus subsequent analysis is straightforward. For instance, one may directly study the worm undulations (Fig. 5f) or extract the worm centerline angle $\psi = \arctan(y(s,t) - y_0(t), x(s,t) - x_0(t))$ to provide insight into the movement patterns and kinematics of the worm (Fig. 5e).

One of the key advantages of our methods is its ability to collect a larger number of samples compared to traditional techniques, while still obtaining reliable results. As the standard error decreases with the number of samples, using our methods allows for metrics to be gathered with less uncertainty while still requiring the same experimental setup. For instance, Fig. 5g shows how the error of estimating the average speed of the center of mass of the nematodes decreases with density. This advantage can be extended to tracking large numbers of nematodes in crowded environments, such as extremely dense petri dishes where more than 6000 concurrent tracks can be simultaneously computed (Fig. 5h). Thus, with our method, we are able to collect a larger number of samples and obtain more precise and reliable results, even in challenging conditions.

## Discussion

We have introduced a deep learning approach for detecting and tracking slender bodies, such as swimming nematodes, in microscopy data. The presented convolutional neural network architecture is capable of accurately detecting a large number of overlapping organisms, a task that can be particularly challenging for standard methods such as bounding boxes and pixel-level classifiers due to the issue of occlusion and overlap. To address this, we have implemented a latent space encoding which allows us to filter by non-maximum suppression and effectively handle overlapping objects. Not only is our method capable of accurately detecting and tracking slender bodies, but it also demonstrates strong scalability, performing well across a range of input frame sizes and densities of bodies. This makes it an ideal tool for a variety of experimental settings where centerlines are useful descriptors, including studies of swimming nematodes, swimming spermatozoa and beating eukaryotic or prokaryotic flagella.

Besides a suitable detector model, labeled training data is also needed. We have demonstrated that relying on a physics-based model to generate synthetic data is adequate to train our network to perform well on real data. This is a key achievement as it means that applications of our system for different experimental studies do not require large datasets to be procured, but rather the implementation of a suitable simulation. Our approach for synthetic data generation relies on over-sampling the behavior of the worms. This is naturally a trade-off as too extreme behavior can lead to datasets that are too hard for the neural network to replicate. For our model, we found that we slightly undersampled certain worm shapes such as strong coiling, which the model therefore could struggle with identifying. Though we did not look into this here, an interesting avenue for future research would be to bootstrap synthetic motility models on small datasets of real organisms. In a similar fashion, the frame-generator procedure should oversample the textures, pixel intensities and noise of real videos. Here, it could be interesting to study whether style transfer[15] or diffusion models[97] could be used to further reduce the gap between training and inference data. We note that we

have only developed and studied a simulation of swimming *C. elegans* worms, and the study of other slender-body systems with our framework requires corresponding synthetic models.

For tracking, we introduced a directed metric that employs past and future centerline predictions to link them across time. At very high densities this may still fail, in particular because the directed metric yields little advantage if predictions are missing in some frames. A potential way to improve on this could come from utilizing the latent space encoding as well. This would require temporal continuity in the latent space representation, which is achievable by modifying the associated loss function. This should enhance the integrity of tracking, as it could potentially be used to resolve issues such as switches by leveraging the separation of close physical predictions with different temporal behavior that characterises the latent encoding. We believe that these suggestions might be fruitful avenues for further research for improving deep learning models for dense detection of centerlines. Furthermore, we note that high short-time scale tracking integrities can still, over longer times, lead to loss of identity. The tracking integrity measurement thus sets the time scale over which accurate statistics can be formed. For longer times scales, other methods are needed[98].

Our approach differs significantly from previous approaches to slender-body tracking. For *C. elegans* tracking in particular, previous trackers have focused on either accurate single worm tracking[95], few worm tracking[39,45,85–89], or large-scale tracking[40,90,91]. However, we found that none of these existing approaches were designed to handle the type of data and densities that we have presented here, and we thus omit quantitative comparisons.

In this paper, we have proposed a new approach for fast and precise detection and tracking of slender bodies in microscopy data. Its speed and accurate performance across a range of densities and sizes, combined with the ability to handle overlapping objects, make it a valuable tool for a variety of experimental settings where precise tracking is essential for obtaining quantitative metrics.

## Methods

### Model structure

*Centerline predictions.* We choose to represent the centre-line of the slender bodies of interest by arrays consisting of $k$ equidistant points (Fig. 2d). These coordinate arrays, which we refer to as centerline, become high-precision descriptors even for complex shapes when $k$ is chosen large. To reduce the complexity of predicting $k$ points, we embed the centerline representation with a principal component (PCA) transform $\mathbf{A}$, the dimension $\kappa$ of which can be much smaller than $k$[50]. The PCA components $\lambda$ represent shape, and in addition hereto, the network also predicts the offset $x_0$ of the centerline, the internal calculation of which is done in a local coordinate system defined by the anchor points. Thus, instead of predicting $2k$ floating point values per centerline, the network needs only output $\kappa + 2$.

The temporal context of the input image stack permits output centerline prediction also for the non-central images. In our approach, we predict a set of three centerlines $z = [x^-, x, x^+]$ corresponding to the three central frames $[I^-, I, I^+]$ of the input stack (Fig. 2c). We consider the middle centerline $x$ the main output, whereas the past $x^-$ and future $x^+$ centerlines are considered auxiliary predictions whose main purpose lies in their use during the latent space encoding as well as the tracking procedure.

We define the similarity measure between two centerlines by the standard Euclidean distance. In the case of detections that look symmetric from either end, we exploit this symmetry and employ the flip-invariant distance defined by

$$d^2(\boldsymbol{x}, \boldsymbol{x}') = \min\left[\sum_{i=1}^{k}(x_i - x_i')^2, \sum_{i=1}^{k}(x_i - x_{k-i+1}')^2\right], \quad (2)$$

as illustrated in Fig. 2e.

Likewise, we define a distance between two collections of consecutive centerlines $z$, $z'$ by their weighted average $d_s^2 = \sum_t \omega_t d^2(z_t, z_t')$, where the weights can be adjusted to give focus to central predictions, and for the present case we choose $\omega = 2\omega^- = 2\omega^+$.

The neural network is trained to minimize the distance $d_s^2$ between predictions and labels. To do so, we let the independent predictors specialize for different shapes. This is achieved by using a permutation-invariant loss such that the total

loss is computed as a sum over the labels only, each using the predictor that best match the labels. Thus many centerline predictions will not contribute to the detection loss.

*Confidence scores.* Each independent prediction of the network includes a confidence score $s$, which is used to filter out bad candidates. In bounding box or mask detection, intersection over union (IoU) is commonly used to evaluate the accuracy of a prediction, however, this metric does not generalize well to centerline predictions when there is overlap. Instead, we introduce a custom metric to define the goodness of a centerline set $z$ by comparing it to its label $\hat{z}$,

$$\hat{s} = \exp\left(-d_s^2(z,\hat{z})/\sigma_s^2\right). \tag{3}$$

Here, $\sigma_s$ is a parameter that sets the scale over which the score varies. The metric is sensitive to perturbations on accurate predictions, i.e. predictions close to labels where $d_s \rightarrow 0$, but loses sensitivity the worse the predictions are. This is a useful feature as correct scoring for good predictions is crucial for choosing the best one, whereas low-scoring predictions are discarded in any case and their relative scoring therefore unimportant.

The score prediction is trained using L2 loss. To avoid conflicting backwards error propagation between this task and that of centerline prediction (as scoring bad predictions is easier), we stop the gradient flow in the computational graph on the last layer of the score-predicting part of $f_\theta$ (Fig. 2a) such that it does not interfere with the accuracy of the predicted centerlines.

*Latent space for candidates suppression.* Finally, we need to ensure that there is only one prediction per object. Bounding box detectors let the user decide the fraction of overlap between prediction boxes of the same class that should be considered to be targeting the same object. As our method must work at high densities, this task is complicated by the fact that two predictions might be very close, even completely overlapping in the central frame, and yet represent different objects. The task of choosing a suitable cutoff distance is therefore difficult, and we make this a trainable task. We do so by embedding each prediction in a low-dimensional latent space in which comparison between predictions is cheap, thus allowing efficient and fast candidate suppression also at high densities.

Our method computes the latent vectors $p$ for predictions using an auxiliary neural network, $q_\phi$ which acts directly on the eigenvalues $\lambda$ and offsets $x_0$ rather than the more redundant centerline coordinate points. We induce a Euclidean metric on the latent space with the interpretation that two predictions $i, j$ are predicting the same object with probability

$$\mathbb{P}(i \leftrightarrow j) = \begin{cases} \exp\left(-||p_i - p_j||^2\right) & \text{if } ||x_{0i} - x_{0j}|| \le \sigma_l, \\ 0 & \text{otherwise.} \end{cases} \tag{4}$$

Here, $\sigma_l$ is a real-space visibility cutoff that prevents far predictions to interact in the encoded space, thus avoiding the need to scale the dimensionality of the latent space with the number of candidates or the input size. We note that when using the flip-invariant metric $d_s$ on centerlines, we explicitly construct the latent space encoder to likewise be flip-invariant.

To train the latent space, we assume that during training predictors are 'trying' to predict the label closest to the prediction centerline. Combined with the probability interpretation, this allows us to use binary cross entropy as a loss function for the probability defined in Eq. (4). To avoid wrong clustering between undefined close-by predictions, the loss contribution of each prediction is scaled by the product of their real scores $\hat{s}_i \hat{s}_j$, thus ensuring that the network focuses its attention on good predictions that will not be filtered out. Finally, since the encoder should not alter the performance of the centerline suggestions, the loss on the latent space representations only updates the weights $q_\phi$ of the encoder, but is trained concurrently with the main model.

We employ non-max suppression to choose the best prediction of each object, but with distances measured in latent space, as illustrated in Fig. 2b. Concretely: Once all the predictions whose score is lower than a threshold $\tau_s$ have been discarded, multiple candidates are likely to still remain for each target object. The lack of low score predictions exposes clusters in the latent space that correspond to single objects. We sort the remaining predictions by their score, automatically accepting the highest-scored one. Once a prediction $i$ is accepted, all predictions $j$ that have a high probability $\mathbb{P}(i \leftrightarrow j) > \tau_o$ of being the same object are removed. This is equivalent to setting an exclusion radius $r_l$ in the latent space as shown in Fig. 2b. We keep iterating on the remaining predictions, pruning the latent space until all candidates have been iterated. The final number of accepted predictions should equal the number of objects in the frame.

## Neural network architecture
*Convolutional neural network.* Most of the weights of the network are at the feature detection convolutional network whose backbone is made of four ResNet groups consisting of 2, 4, 4, 2 blocks with strides 1, 2, 1, 2, respectively. We modify the original ResNet architecture by replacing the initial max-pooling layer with an average-pool layer to avoid translational invariance. The final shape of the feature space is $[H/16, W/16, C]$, with $C$ being the number of candidates each cell proposes. We have set $C = 8$ for this project in order to fulfill the condition of the number of

predictions being larger than the number of bodies even at high densities. All in all, there will always be $C$ candidates per cell regardless of input size, which leads to a large number of candidates to be sorted in the filtering process. The head of the convolutional neural network is composed of two fully connected layers of 512 and $C \cdot (3(m + 2) + 1)$ cells, respectively, with batch normalization in between. Due to the orientation invariance of the loss function on the centerline predictions, it is possible that the centerlines in the predicted set $x^-, x, x^+$ are not aligned. To remedy this, we aligned them by comparing them with the eigenvalues of the flipped centerline. In order to get the *flipped eigenvalues* $\lambda_f$, we use

$$\lambda_f = \mathbf{A}^{-1}\mathbf{J}\mathbf{A}\lambda \tag{5}$$

where $\mathbf{A}$ is the PCA transformation matrix and $\mathbf{J}$ is the exchange matrix.

*Latent space encoder.* The encoder $q_\phi$ is composed of two fully connected layers with batch normalization in-between. The input of the encoder is the vector of size $3(m + 2)$ characterizing the centerline predictions and the output is $D$ floating point values, corresponding to the coordinates of $p$ in the $D$-dimensional latent space. We have found $D = 8$ to be a well-performing dimension in our experiments. Due to the orientation invariance of the centerlines predictions, we need to construct the encoder to cluster those centerlines regardless of orientations as well. To do so, the input values are expanded to include those of the flipped centerlines $\lambda \rightarrow (\lambda, \lambda_f)$ and both are fed to the same layer. To ensure symmetry, the output is then summed before passing through the last layer. In doing so, the encoder becomes independent of centerline orientation.

## Training
*Simulation-based training.* Our in silico data generator has two main components: a physics-based model for the organism and a synthetic frame generator.

In silico worms are generated on demand every training step which removes the possibility of overfitting to the generated frames. In order to train the model to work effectively with a range of worm densities, we generate batches with different numbers of worms in a uniform manner, without bias towards low or high worm counts. This teaches the model to handle a variety of densities without overfitting to any specific case. And to make the model more robust, training also happens on densities whose manual annotation would be extremely challenging. The simulation and video synthesis are implemented in a GPU framework which enables fast end-to-end training without the performance penalization of data transferring between the accelerator and the host machine.

We base the worm simulation on resistive force theory, as it has previously been shown to correctly predict the position of the skeleton for short spans of time[99]. Since the network only perceives the frames surrounding the target frames, we found the total duration of the clip to be short enough that a linear swimming model approximation fits our needs. The physics-based model should encapsulate all types of organism behavior. This can be achieved by oversampling the behavior, i.e. by making the simulations more diverse in the behavior than reality and thus hope to include all types of real behavior as well. Details on the worm simulation and video synthesis can be found in the in silico dataset section of the methods.

Despite the potential for physics-based simulations to be used for synthetic training data, discrepancies with real data may lead to inaccuracies when applied to real microscopy images. This reality gap can be the result of an overly simplified motility model or physics model, or the result of imprecise video synthesis. The gap may be further increased by the fact that the model relies on the PCA transformation matrix $\mathbf{A}$ obtained on synthetic data, where the number of PCA components used have been chosen to accurately reproduce all synthetic patterns, but not necessarily to generalize to out-of-sample videos. Thus we find that our model is limited to accurate skeleton predictions only on shapes that resemble those produced by our simulations, and the goal of the simulations is therefore to reproduce a broad spectrum of possible motility patterns. Likewise, we find that our model is susceptible to the brightness of the videos, and accordingly we adjust the real videos to increase their resemblance to the training data.

*Loss functions.* Centerline descriptors are trained as a regression problem. Thus, the loss contribution is given by the custom distance defined in Eq. (2). To enforce specialization on the predictors, and due to the number of predictions $M$ being considerably larger than the number of bodies $N$, only the best predictors are accounted for in the loss. Nevertheless, there may be labels $\hat{x}$ completely or partially outside the frame at $t_c$, despite being inside at $t_0$. To make sure not to punish bad predictions at the boundaries for not matching *invisible* centerlines, instead of using the number of simulated bodies $N$, the subset of bodies completely inside the frame $N_v$ is used and the final loss expression is given by:

$$l_x = \frac{1}{N_v}\sum_i^{N_v} \min_m d_s^2(z_m, \hat{z}_i) \tag{6}$$

The score L2 loss is computed as the difference between the values predicted and the score the centerline proposals should have. Thus, using Eq. (3), we train the predicted score of all predictions using:

$$l_s = \frac{1}{M}\sum_i^M \left(\exp\left(-\min_n \frac{d_s^2(z_i, \hat{z}_n)}{\sigma_s}\right) - s\right)^2 \tag{7}$$

Finally, the loss function for the latent space encoder is a modified cross entropy loss scaled by the product of scores. Denote $\mathbb{P}_{i,j} = \mathbb{P}(i \leftrightarrow j)$ as defined in Eq. (4), then the encoder loss is defined as an average over all pairs of predictions $\langle i, j \rangle$ that are physically within the cutoff $\sigma_l$,

$$l_p = \frac{1}{S} \left\langle \hat{s}_i \hat{s}_j (t_{ij} \log(\mathbb{P}_{i,j}) + (1 - t_{i,j}) \log(1 - \mathbb{P}_{i,j})) \right\rangle_{\langle i,j \rangle}, \quad (8)$$

where $S = \sum \hat{s}_i \hat{s}_j$, and $t_{i,j}$ indicates whether $i$ and $j$ are targeting the same label $k$, and is set by

$$t_{ij} = \begin{cases} 1 & \text{if } k_i = k_j \\ 0 & \text{otherwise} \end{cases} \quad (9)$$

with $k_i, k_j$ being the closest labels to the predictions $z_i, z_j$ respectively.

*Training details.* Training has been done from scratch, i.e. without the use of a pretrained backbone. During training, the frame size for the input clips used was $256 \times 256$, but due to the anchored approach, this does not constrain inference to happen at the same resolution. Synthetic input is generated on demand and on device rather than using a fixed pre-generated dataset. Thus, the network never sees the same frame twice and there is no host-to-device data transfer. As mentioned in the main text, all networks are trained simultaneously, despite the weights of each one depending on different cost functions. The code has been written in JAX using HAIKU and training has been carried out on a cluster of $8 \times$ NVIDIA A5000's.

**Inference**. Inference happens at any resolution whose dimensions are multiple of 16. The input frames need to be slightly pre-processed as mentioned in the previous sections. Candidate predictions are chosen using a score threshold, and non-maximum suppression in latent space is used for filtering. Due to the sequential nature of the filtering process, the implementation is written to use the CPU using NUMBA.

*Input clips pre-processing.* The images used to train the model have dark (small pixel intensity) backgrounds, as we employ zero-padded convolutional layers. This is relevant for real recordings, where a negative flip may be necessary to match the network requirements. During training, generated clips are normalized using a 1–99 percentile normalization. For real clips, to accommodate uneven lighting conditions and potential obstructions we apply contrast limited adaptive histogram equalization (CLAHE) and subsequently correct the intensity of videos to match the variations of the simulated data (see Supplementary Fig. 1). Note that we match real data to the synthetic as this avoids the need to retrain the network for different experimental setups.

**In silico dataset**
*Worm simulation.* Worm trajectories are computed by employing a resistive force theory swimming model used to predict rigid body motions of *C. elegans* from the undulations[99]. Thus, we ensure that from a given set of generated undulations, the produced motions will match those of real worms. From empirical observations, we propose a simple Eq. (10) to generate the undulation of swimming adult worms. The set of equations is specifically targeted to the dataset of dense swimming *C. elegans*, but we expect it to also apply to other life stages by modifying the parameters of the sampling distributions. Similarly, the undulations proposed do not take into account self-coiling, as it is rare on free swimming nematodes, but changing Eq. (10) appropriately would allow the system to learn to detect them. We define the motions by the centerline angle $\psi_k(s)$ with $s \in [0, 1]$ [Fig. 2d], and decompose this into a linear combination:

$$\psi(s) = \psi_u(s, t) + \psi_s(s, t). \quad (10)$$

This logically separates the worm undulations into two types of motion: one corresponding to a sinusoidal motion $\psi_s$ and one in which the whole body bends $\psi_u$. These we define by

$$\psi_u(s, t) = A \cos\left(\frac{2\pi}{T} t + \rho_1\right) \cos\left(k_u s_k + \rho_2\right) \quad (11)$$

$$\psi_s(s, t) = \tilde{A} \cos\left(\frac{2\pi}{T} t + k_s s_k + \rho_3\right) \quad (12)$$

where $\tilde{A} = \frac{1}{2}(1 + |\sin(2\pi t)|)A$ and the rest of parameters are sampled from random distributions. Although many improvements for the above equations can be suggested, we prefer to keep the model simple.

Once the values of the parameters for $\psi$ are generated for all the timesteps of the simulation, the positional coordinates are obtained using

$$\vec{x}(s, t) = L \int_0^s \begin{pmatrix} \cos(\psi(s', t) + \gamma) \\ \sin(\psi(s', t) + \gamma) \end{pmatrix} ds' \quad (13)$$

where $\gamma$ is a random orientation and $L$ is the length of the worm (also sampled).

Once the skeleton is defined, the rigid body motions are predicted by solving[99]

$$\vec{F} = \int_0^L \vec{f} \, ds = 0, \quad (14)$$

$$\vec{\tau} = \int_0^L (\vec{x} - \vec{x}_{\text{CoM}}) \times \vec{f} \, ds = 0, \quad (15)$$

where the force $\vec{f}$ can be calculated from the centerline velocity $\vec{U} = \partial_t \vec{x} + V + \Omega \times (\vec{x} - \vec{x}_{\text{CoM}})$ by

$$\vec{f} = \alpha_t (\hat{t} \cdot \vec{U}) \hat{t} + \alpha_n (\hat{n} \cdot \vec{U}) \hat{n}. \quad (16)$$

Here, $V$ and $\Omega$ are the center-of-mass velocity and rotational velocity (that we are solving for), and $\alpha_t$ and $\alpha_n = \alpha \alpha_t$ is the tangential and normal drag coefficients, which is also sampled for $(\alpha > 1)$. We did not find a need for using a non-linear force theory. The simulation is run with Python 3.9 using the JAX library.

*Video synthesis.* Given the labels for the centerlines positions, synthetic videos are generated to be used as input during training. In order to add width to each worm, we vary the local body radius $r$ by a function of the form

$$r(s) = \tilde{R} |\sin(\arccos(as + b))| \quad (17)$$

The pixel values of those circles are calculated with anti-aliasing. Once the worms have been rendered, noise artefacts such as uneven background, blurring, Gaussian noise, etc. are added to replicate the observed conditions of real experiments. During training, standard augmentation techniques are applied as well. In the same manner as the simulation of the motion and the neural network training, frame generation is also written in Python using the JAX library in order to leverage GPU capabilities.

**Evaluation**
*Experimental dataset.* Videos of swimming *C. elegans* were filmed using the protocol described in ref. [40].

*Manually annotated dataset.* The evaluation dataset is annotated using a custom tool that can be found at https://github.com/kirkegaardlab/deeptanglelabel. Around ~1500 centerlines have been annotated (see data availability).

*Asymmetric dynamic time-warped error metric.* We introduce a custom metric to suitably compare the densely defined centerlines of the predictions to labels that are defined using only a few labeled points. The metric used must be shift-invariant, as having points anywhere along the centerline should yield zero error regardless of whether the label points precisely coincide with the prediction points or not. Likewise, label points should be monotonically assigned along the centerline in order to avoid artificially reducing the error for strongly bent or self-coiling worms. Finally, it must be robust against the subjectivity of the labellers, as manual annotations might miss or avoid spots where visibility is low such as the end-points of the worms.

To satisfy all these requirements, we introduce a metric based on the dynamical time warping (DTW) distance used to measure the similarity between temporal curves. In our modified version, asymmetric DTW, summation only runs over label points. Thus, the metric $\delta_{\text{adtw}}$ is defined as follows: Let $d(i, j)$ be the Euclidean distance between label point $i$ and prediction line segment $j$, then

$$\delta_{\text{adtw}} = \min_\alpha \frac{1}{N} \sum_{i=1}^N d(i, \alpha(i)), \quad (18)$$

where $\alpha: [1, N] \to [1, M]$ is a monotonic (non-decreasing or non-increasing) assignment of the $N$ label points to the $M$ prediction line segments. A visual representation of the metric is shown in Fig. 4b, and the $\mathcal{O}(NM)$ algorithm for its calculation is detailed in Table. 1. We note that, just as is the case for the dynamic time warping distance, this is not a true distance in the mathematical sense.

**Tracking**
*Tracking implementation.* Tracking is done by sequentially predicting individual frames. For better performance, batching of frames allows for parallel detections and can drastically reduce execution time. Nevertheless, due to the requirement of including surrounding frames for each detection, a considerable increase in memory usage is observed. Once a collection of centerline detections is obtained, each prediction is adapted in order to make it work with the TrackPy Python library. Due to the peculiarity of our distance metric, we implement a custom neighbor strategy that avoids the assumption of a symmetric distance function.

It may happen that some detection artefacts appear during the sequential detection performed during tracking. We have implemented a quick check on the resulting tracks to make sure not to have stubs, and fix obvious branching of tracks due to these artefacts. A slight increase in integrity is observed on dense clips.

*Tracking integrity.* Given a true label of a track of length $N$, we associate to this track at each time point $i$ a prediction identity $I_i$. We may then define the integrity

**Table 1 Algorithm for asymmetric dynamic time warping.**

**Data:** Label curve defined by $N$ points $\{p_i\}$, and prediction curve defined by $M$ line segments $\{s_j\}$.
**Result:** The asymmetric dynamically time-warped distance from label to prediction.

Initialize matrices $C, D$ with size $[N, M]$.
**for** $i = 1$ **to** $N$ **do**
 **for** $j = 1$ **to** $M$ **do**
  $D_{i,j} \leftarrow$ `distance_from_point_to_segment`$(p_i, s_j)$
$C_{1,1} \leftarrow D_{1,1}$
**for** $i = 2$ **to** $N$ **do**
 $C_{i,1} = C_{i-1,1} + D_{i,1}$
**for** $j = 2$ **to** $M$ **do**
 $C_{1,j} = \min(C_{1,j-1}, D_{1,j-1})$
**for** $i = 2$ **to** $N$ **do**
 **for** $j = 2$ **to** $M$ **do**
  $C_{i,j} = \min(C_{i,j-1}, C_{i-1,j} + D_{i,j})$
**return** $C_{N,M}/N$

of the track as $\iota = 1/N^2 \sum_{i=1}^{N} \sum_{j=1}^{N} [I_i = I_j]$. For instance, if a label is given identities $I = [1, 1, 1, 5, 5, 5, 3, 3, 3]$ during the track, i.e. there have been two identity swaps, we find $\iota = \frac{1}{3}$, which has the interpretation that the track was correct for a third of the time. This measure will in general scale like $\iota \sim N^{-1}$, as longer tracks will have higher likelihood of identity swaps.

**Reporting summary**. Further information on research design is available in the Nature Portfolio Reporting Summary linked to this article.

## Data availability
Dataset videos and labels can be found at https://zenodo.org/record/8093305. Pretrained weights can be downloaded at https://sid.erda.dk/share_redirect/FEUsqYEqBy. The source data for graphs and charts is available as Supplementary Data 1 and any remaining information can be obtained from the corresponding author upon request.

## Code availability
Main code can be found at https://github.com/kirkegaardlab/deeptangle with a fixed version available at https://zenodo.org/record/8093334. and code for labeling videos at https://github.com/kirkegaardlab/deeptanglelabel.

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

## Acknowledgements

Videos of *C. elegans* were provided by Celia Raimondi, Sunehera Sarwat and Michele Perni. This work was supported by the Novo Nordisk Foundation, Grant Agreement NNF20OC0062047.

## Author contributions

A.A. and J.B.K. designed research, A.A. performed research, A.A. and J.B.K. analyzed results, A.A. and J.B.K wrote the paper.

## Competing interests

The authors declare no competing interests.
