## [Peer Review File · Communications Biology]

Reviewers' comments:

Reviewer #1 (Remarks to the Author):

"Fast spline detection in high density microscopy data" by Alonso and Kirkegaard presents a new worm tracking approach that is fast and finds accurate midlines even with overlapping worms. This is an impressive paper and is an exciting advance for the field. It is much more than "another CNN applied to segmentation/tracking" and includes conceptual advances to make the system work. I think it can be published with only minor (but I feel important) modifications.

There are a few places where I think wording should be changed to make it clearer what has and hasn't been demonstrated in this paper. The authors write in several places about the application of the system to tracking other slender objects but they have only demonstrated their algorithm on worms. For example, in the introduction "This is applicable to a broad class of systems [Fig. 1], including tracking of nematode worms [48–50], spiral or elongated bacteria [51–54], spermatozoa [55, 56], the flagella of both eukaryotes [42, 43] and prokaryotes [57], and freely swimming flagella such those of microgametes [58]."

This is too strong a statement given that they have only shown that it works on swimming worms. I agree that in principle there is nothing worm-specific about the algorithm, but there are some possible barriers to generalisation including the requirement to develop a reasonable physical model to simulate training data. It is also not at all clear how the algorithm would handle more complicated kinds of overlaps (e.g. longer objects with more shape degrees of freedom that get more entangled).

Related to this, I would change all the references to crawling *C. elegans* to swimming *C. elegans*. Swimming is more stereotyped, worms adopt less complex shapes in liquid, and they move more independently when the overlap compared to crawling worms. The performance on crawling worms therefore could be quite different.

There are several places where I think the citations could be more representative. In the second paragraph, the authors cite the LEAP paper on pose estimation. Since they're talking about multiple animals in the paragraph, the more recent multi-animal version would make more sense. <https://www.nature.com/articles/s41592-022-01426-1>

When discussing previous skeletonizing trackers, they cite Rizvandi to argue previous overlap approaches require ad-hoc procedures, but they don't cite the model-based methods by Fontaine (<https://ieeexplore.ieee.org/document/4462606>) or Wahlby (<https://www.nature.com/articles/nmeth.1984>).

The paper (including the title) talks about splines, but the authors state "We represent the splines by k equidistant points along the centreline of the body". Splines aren't collections of points so I think it would be clearer if references to splines were replaced with a more appropriate term like

centerline or perhaps skeleton.

On page 5 the authors emphasise the advantages of simulated training data (which I agree with!) but it would probably be good to mention limitations as well. Specifically, it only works on systems where you can simulate the appropriate behaviour. That won't be the case for many objects of interest.

Page 9: The tracking integrity sounds good initially, but even high accuracy locally can quickly lead to randomised identities across a population. For example, in Perez-Escudero et al. (<https://www.nature.com/articles/nmeth.2994>) they show that even with 99% of crossings solved correctly, only 11% of assignments are correct after 2 minutes of simulated zebrafish tracking. For some applications, shorter high-confidence tracks might be more desirable than longer tracks with ID switches. Is there a way of using tracking confidence to avoid switching errors by breaking tracks?

For this reason, the last paragraph before the discussion could be made more nuanced. I agree with the advantage of estimating something like centroid speed of the population, but if there is significant track mixing then inter-individual variation would get averaged out (e.g. if there are two subgroups A and B, after a long video, most tracks will incorrectly contain data from both A and B animals so a track-level average might mask inter-individual variability).

On page 10 when discussing the PCA representation and the fact that k can be much smaller than n , it would be good to cite the a demonstration of this for worms (Stephens et al. <https://journals.plos.org/ploscompbiol/article?id=10.1371/journal.pcbi.1000028>)

Because the flip-invariant distance is used, good matches are found regardless of flips, but are the final outputs oriented consistently (i.e. is the first point always the head of the worm)? If not, this should be stated explicitly. If so, it would be good to know how this was done.

On page 14 the authors state that "we match real data to the synthetic as this avoids the need to retrain the network for different experimental setups". I can see the point in principle, but this should actually be demonstrated.

Minor points

-page 1: "sensible to noise" -> "sensitive to noise"

-page 2: "is principally applicable" -> "is in-principle applicable"

-page 12: when $M \gg N$ is used, I don't think they have yet been defined (although they are defined below)

-page 13: the 3rd paragraph of the training section refers to details of the video synthesis being in the methods, but this is already the methods section so best to be more specific.

-page 14: "as described in the previous sections" -> "as mentioned in the previous sections" (they are not described until the next paragraph)

-page 15: "all for the timestamps" -> "for all the timestamps"

Reviewer #2 (Remarks to the Author):

The manuscript titled "Fast spline detection in high density microscopy data" by Albert Alonso and Julius B. Kirkegaard presents a novel deep learning approach for estimating the postures of *C. elegans* worms as spline curves. The work demonstrates the ability to identify and track thousands of worms in dense environments (up to 2-3 worms/mm²) with minimal error rate. They do this by generating synthetic data from a numerical model to train their deep learning model, which I find to be an innovative approach.

It is an interesting and relevant work and I believe it can potentially contribute significantly to the field of pose estimation of slender organisms. There are a few questions, and some minor points that I would like to ask the authors, listed below:

Major Questions:

- 1) Can the authors compare their proposed algorithm with pre-existing methods for worm segmentation, like Tierpsy or Wormpose using some normalized metric? The manuscript would greatly benefit from such a comparative analysis.
- 2) The authors should provide more information on the robustness of their model to variations in lighting conditions, imaging modalities, or the presence of debris in the experimental setup. Perhaps an image in the supplementary material demonstrating the cases where the segmentation fails due to varied recording conditions.
- 3) Does synthetic data generation incorporate movement types across life stages? Maybe the authors could comment on whether their model can effectively estimate the postures and track the movements of worms at different life stages or if modifications are needed to accommodate these variations.
- 4) If synthetic data for self-overlapping postures like the Greek letter "alpha" could be generated, is there a possibility for detecting self-intersecting worm's postures? This can be useful in estimating the posture of worms with longer aspect ratios and which show more complex posture topologies.
- 5) How long was the model trained to obtain the accuracy observed in this work? I believe that would be a relevant measure to understand the feasibility of implementing this approach in a differently behaving filamentous object/organism.

6) As a follow-up to my previous point, it would be interesting to know if the authors have explored or considered using transfer learning techniques to adapt their model to other filamentous organisms more efficiently.

Minor Points:

1) The sentence "We found that humans are better at correctly resolving overlap between moving bodies when given access to videos rather than still micrographs." does not have a reference or a justification. Can the authors clarify the reasoning behind this statement?

2) On page 5, first paragraph, the worm density is given units mm^{-1} . I believe it should be mm^{-2} . Please correct this error.

3) In the methods section on page 13, there is a sentence saying, "Details on the worm simulation and video synthesis can be found in the methods section." The sentence refers to the method section even though it is within the methods section. Please revise this sentence for clarity.

Best Regards,
Ishant Tiwari
School of Chemical and Biomolecular Engineering,
Georgia Tech

Response to Referee Reports

Fast spline detection in high density microscopy data

Albert Alonso & Julius B. Kirkegaard
Nature Communications Biology,

Reviewers' Comment, Authors' Response, Manuscript Text

Response to Report of Referee 1

"Fast spline detection in high density microscopy data" by Alonso and Kirkegaard presents a new worm tracking approach that is fast and finds accurate midlines even with overlapping worms. This is an impressive paper and is an exciting advance for the field. It is much more than "another CNN applied to segmentation/tracking" and includes conceptual advances to make the system work. I think it can be published with only minor (but I feel important) modifications.

Thank you. We are pleased to hear the feedback.

There are a few places where I think wording should be changed to make it clearer what has and hasn't been demonstrated in this paper. The authors write in several places about the application of the system to tracking other slender objects but they have only demonstrated their algorithm on worms. For example, in the introduction "This is applicable to a broad class of systems [Fig. 1], including tracking of nematode worms [48–50], spiral or elongated bacteria [51–54], spermatozoa [55, 56], the flagella of both eukaryotes [42, 43] and prokaryotes [57], and freely swimming flagella such those of microgametes [58]."

This is too strong a statement given that they have only shown that it works on swimming worms. I agree that in principle there is nothing worm-specific about the algorithm, but there are some possible barriers to generalisation including the requirement to develop a reasonable physical model to simulate training data. It is also not at all clear how the algorithm would handle more complicated kinds of overlaps (e.g. longer objects with more shape degrees of freedom that get more entangled).

We agree with the reviewer on this point, and have turned down the language appropriately on the following snippets of text. Text in bold represents text added for clarification.

(Page 1)

Our method enables both accurate shape prediction and tracking in dense experiments of slender objects, a key challenge for a broad class of systems [Fig. 1], including tracking of nematode worms [48–50], spiral or elongated bacteria [51–54], spermatozoa [55, 56], the flagella of both eukaryotes [42, 43] and prokaryotes [57], and freely swimming flagella such those of microgametes [58].

(Page 2)

Our method is in-principle applicable to all microscopy datasets that involve slender bodies, **but we do not develop its general applicability here. Instead, we focus on ...**

We have further added to the discussion:

(page 10)

We note that we have only developed and studied a simulation of swimming *C. elegans* worms, and the study of other slender-body systems with our framework requires corresponding synthetic models.

*Related to this, I would change all the references to crawling *C. elegans* to swimming *C. elegans*. Swimming is more stereotyped, worms adopt less complex shapes in liquid, and they move more independently when the overlap compared to crawling worms. The performance on crawling worms therefore could be quite different.*

We have replaced all the references of crawling to swimming, as we agree it better describes the targeted dataset. Thank you for the comment.

There are several places where I think the citations could be more representative. In the second paragraph, the authors cite the LEAP paper on pose estimation. Since they're talking about multiple animals in the paragraph, the more recent multi-animal version would make more sense. <https://www.nature.com/articles/s41592-022-01426-1>

We have updated the citation to reference the SLEAP paper, as we agree to be more relevant to our paper than its predecessor.

When discussing previous skeletonizing trackers, they cite Rizvandi to argue previous overlap approaches require ad-hoc procedures, but they don't cite the model-based methods by Fontaine (<https://ieeexplore.ieee.org/document/4462606>) or Wahlby (<https://www.nature.com/articles/nmeth.1984>).

We now include these references as well:

This problem has traditionally been approached by employing pixel-wise segmentation and subsequent skeletonization procedures [38–43], an approach that requires **model-based approaches [44, 45]** or ad-hoc procedures [46] to solve the problem of correctly identifying overlapping organisms, the combinatorial complexity of which blows up at high densities.

The paper (including the title) talks about splines, but the authors state "We represent the splines by k equidistant points along the centreline of the body". Splines aren't collections of points so I think it would clearer if references to splines were replaced with a more appropriate term like centerline or perhaps skeleton.

We agree with the reviewer that the word *spline* is technically incorrect. We felt, however, that it made for simpler reading and conveyed the smoothness of the predictions. Nonetheless, we have now followed the recommendation of the reviewer. The title of the paper has been changed to "**Fast detection of slender bodies in high density microscopy data**", and we have replaced all occurrences of *splines* by *centerlines*.

On page 5 the authors emphasise the advantages of simulated training data (which I agree with!) but it would probably be good to mention limitations as well. Specifically, it only works on systems where you can simulate the appropriate behaviour. That won't be the case for many objects of interest.

We have included a discussion on the matter of disadvantages of limitations of synthetic datasets:

(Page 5)

Physics-based synthetic datasets have successfully been used to train systems on similar conditions, for instance where manual labelling may introduce unnecessary noise or bias to the model [16]. **Naturally, this requires the formulation of a physical model that is accurate on the relevant time scales. Furthermore, dependence on synthetic datasets can result in a divergence between the target and the training data, potentially leading to inaccurate predictions during inference — a phenomenon that is avoided if the model is trained on real data.**

Page 9: The tracking integrity sounds good initially, but even high accuracy locally can quickly lead to randomised identities across a population. For example, in Perez-Escudero et al. (<https://www.nature.com/articles/nmeth.2994>) they show that even with 99% of crossings solved correctly, only 11% of assignments are correct after 2 minutes of simulated zebrafish tracking. For some applications, shorter high-confidence tracks might be more desirable than longer tracks with ID switches. Is there a way of using tracking confidence to avoid switching errors by breaking tracks?

For this reason, the last paragraph before the discussion could be made more nuanced. I agree with the advantage of estimating something like centroid speed of the population, but if there is significant track mixing then inter-individual variation would get averaged out (e.g. if there are two subgroups A and B, after a long video, most tracks will incorrectly contain data from both A and B animals so a track-level average might mask inter-individual variability).

This is indeed true. We already mention on tracking integrity:

This measure will in general scale like $\iota \sim N^{-1}$, as longer tracks will have higher likelihood of identity swaps.

But have now added a more detailed discussion of this fact:

(page 10)

We note that high short-time scale tracking integrities can still, over longer times, lead to loss of identity. The tracking integrity measurement thus sets the time scale over which accurate statistics can be formed. For longer times scales, other methods are needed [98].

On page 10 when discussing the PCA representation and the fact that kappa can be much smaller than k, it would be good to cite the a demonstration of this for worms (Stephens et al. <https://journals.plos.org/ploscompbiol/article?id=10.1371>).

Thank you, this reference has now also been cited in that sentence.

Because the flip-invariant distance is used, good matches are found regardless of flips, but are the final outputs oriented consistently (i.e. is the first point always the head of the worm)? If not, this should be stated explicitly. If so, it would be good to know how this was done.

We have now included text stating explicitly the alignment of the predictions.

(page 3)

...and the output of our model is composed of a set of centerline predictions $z = [x^-, x, x^+]$ representing the past, present, and future motion of the bodies. We represent the centerlines by k equidistant points along the centreline of the body [Fig. 2d]. **The centerlines contained within the set maintain alignment with a consistent head positioning across the three predictions.**

(page 8)

Notice that due to the flip-invariance of the distance metric, consecutive assignments of sets of centerlines are not necessarily aligned, hence a trivial alignment of the centerlines is carried out during post-processing before being analysed.

On page 14 the authors state that "we match real data to the synthetic as this avoids the need to retrain the network for different experimental setups". I can see the point in principle, but this should actually be demonstrated.

We indeed skipped mentioning this aspect. We have now included a comment on the matter:

(page 15)

For real clips, to accommodate uneven lighting conditions and potential obstructions we apply contrast limited adaptive histogram equalization (CLAHE) and subsequently correct the intensity of videos to match the variations of the simulated data (see SI figure).

Furthermore, we now include an example of the performance of the model on unprocessed input clips [Figure 1, here] as an SI figure. This demonstrates the effect of preprocessing and also to shows the performance of the model near obstructions (such as the air bobbles shown here).

Figure 1

Minor points

-page 1: "sensible to noise" -> "sensitive to noise"

-page 2: "is principally applicable" -> "is in-principle applicable"

- page 12: when $M \gg N$ is used, I don't think they have yet been defined (although they are defined below)

-page 13: the 3rd paragraph of the training section refers to details of the video synthesis being in the methods, but this is already the methods section so best to be more specific.

-page 14: "as described in the previous sections" -> "as mentioned in the previous sections" (they are not described until the next paragraph)

-page 15: "all for the timestamps" -> "for all the timestamps"

Thank you for all the comments, these have all been fixed.

Response to Report of Referee 2

*The manuscript titled "Fast spline detection in high density microscopy data" by Albert Alonso and Julius B. Kirkegaard presents a novel deep learning approach for estimating the postures of *C. elegans* worms as spline curves. The work demonstrates the ability to identify and track thousands of worms in dense environments (up to 2-3 worms/mm²) with minimal error rate. They do this by generating synthetic data from a numerical model to train their deep learning model, which I find to be an innovative approach. It is an interesting and relevant work and I believe it can potentially contribute significantly to the field of pose estimation of slender organisms. There are a few questions, and some minor points that I would like to ask the authors, listed below:*

Thank you for the very constructive and relevant feedback for our manuscript.

Major Questions:

1) Can the authors compare their proposed algorithm with pre-existing methods for worm segmentation, like Tierpsy or Wormpose using some normalized metric? The manuscript would greatly benefit from such a comparative analysis.

We absolutely wanted to include such a comparison. However, after running existing, available software on dense videos of *C. elegans* we found the comparison to previous methods to become too negative/unfair when evaluated on the type of data we consider: the available software is simply not designed to handle overlap at the scale that we consider.

Concretely, this is the best output we were able to get from TIERPSY:

Only a single worm was detected. TIERPSY struggles for two reasons: (1) it was designed for high resolution videos and (2) it is not designed to handle overlap. For instance, running TIERPSY on high-resolution videos, we can detect all singular worms, but overlapping worms are incorrectly merged:

While we could include this as a comparison, we prefer not to. Likewise, it does not make much sense to compare to WORMPOSE, as this software can only do predictions for single worms, and thus inherently cannot handle multi-organism-overlap.

Furthermore, for other worm tracking software, similar problems exist. Particularly, multi-worm trackers such as the 'Kerr Lab Multiwormtracker', the 'Parallel Worm Tracker' and the 'Wide Field-of-View Nematode Tracking Platform' all ignore overlap and thus cannot be used as fair grounds for comparison. For instance, here is the output from the Wide Field-of-View Nematode Tracking Platform:

which shows that overlapping worms get assigned to the same cluster which is then subsequently ignored for analysis. Additionally, those software platforms do not compute worm shapes, but only statistics such as swimming speeds.

We do agree with the reviewer that these considerations should be mentioned in the paper. Thus we now include this as part of the discussion:

Our approach differs significantly from previous approaches to slender-body tracking. For *C. elegans* tracking in particular, previous trackers have focused on either accurate single worm tracking [95], few worm tracking [39, 45, 85–89], or large-scale tracking [40, 90, 91]. However, we found that none of these existing approaches were designed to handle the type of data and densities that we have presented here, and we thus omit quantitative comparisons.

2) The authors should provide more information on the robustness of their model to variations in lighting conditions, imaging modalities, or the presence of debris in the experimental setup. Perhaps an image in the supplementary material demonstrating the cases where the segmentation fails due to varied recording conditions.

We have now included Fig.1 in the SI to exemplify our model’s performance under different lighting conditions and with debris. The figure shows the importance of normalizing the input using CLAHE and intensity correction.

In the 4th frame of the figure, we see a case of double prediction where a single worm gets assigned two detections. This is due to intensity differences between training and post-process input clips. Thus, matching to synthetic data can require fine tuning for perfect results in difficult cases such as this one.

3) Does synthetic data generation incorporate movement types across life stages? Maybe the authors could comment on whether their model can effectively estimate the postures and track the movements of worms at different life stages or if modifications are needed to accommodate these variations.

Our synthetic training setup is simple to tweak and adapt to different situations such as worm life stages. However, in this work we focus on adult, swimming worms only. We now mention this in the manuscript’s methods section about worm simulation.

(page 14)

From empirical observations, we propose a simple equation, Eq. (10), to generate the undulations of **swimming adult worms**. **The set of equations is specifically targeted to the dataset of dense swimming *C. elegans*, but we expect it to also apply to other life stages by modifying the parameters of the sampling distributions. Similarly, the undulations proposed do not take into account self-coiling, as it is rare on free swimming nematodes, but changing Eq. (10) appropriately would allow the system to learn to detect them.**

4) *If synthetic data for self-overlapping postures like the Greek letter “alpha” could be generated, is there a possibility for detecting self-intersecting worm’s postures? This can be useful in estimating the posture of worms with longer aspect ratios and which show more complex posture topologies.*

Absolutely. There is no restriction on the type of spline shapes that can be trained on and predicted as long as the shapes can be encoded by a PCA/-representation. Self-overlapping shapes (such as α) are indeed possible. We now mention this (see paragraph above).

5) *How long was the model trained to obtain the accuracy observed in this work? I believe that would be a relevant measure to understand the feasibility of implementing this approach in a differently behaving filamentous object/organism.*

The duration of training is highly related to the complexity of the problem, e.g. the density of worms. Training from scratch takes around 28 hours on single A5000 GPU in order to achieve similar performance to the one shown in the manuscript. If instead of targeting frames with extreme densities (~ 6.1 worms/mm²) we limit the synthetic simulation to lower densities (~ 1.22 worms/mm²), the training time gets reduced to a couple of hours. These numbers can probably be reduced significantly by optimizations, for instance via hyperparameter search. We have not studied this as we have focused rather on fast *inference* times. We should also note that a majority of this training time is due to synthetic video generation on-the-fly. A more efficient, but less dynamic, approach may come from creating a single, large synthetic dataset and train without the need of generate new samples every train step.

Finally, we should mention that adapting the model to new simulation parameters can easily take advantage of pretrained weights, and the need to train from scratch should be rare.

6) *As a follow-up to my previous point, it would be interesting to know if the authors have explored or considered using transfer learning techniques to adapt their model to other filamentous organisms more efficiently.*

We have not adapted the model to train on other filamentous organisms in this work. However, we have relied successfully on transfer learning to speed up research when new modifications of the model were used. Since it now has become rare in the machine learning community to promote training from scratch instead of relying on transfer learning from foundational models, we strongly believe that new models targeting other slender organisms would benefit considerably from starting with our learned backbone network.

Minor Points:

1) *The sentence “We found that humans are better at correctly resolving overlap between moving bodies when given access to videos rather than still micrographs.” does not have a reference or a justification. Can the*

authors clarify the reasoning behind this statement?

This statement comes from personal experience when manually labelling the data and may be classified as anecdotal evidence. Regardless, we consider that adding such a claim is not strictly necessary in order to justify the need of adding temporal context to the network. Instead, we have changed the manuscript text to be

In still micrographs, the identities of individual worms can be obscured by overlaps making them impossible to accurately identify, and only by relying on the adjacent frames can they be correctly resolved. Thus, to allow ...

2) On page 5, first paragraph, the worm density is given units mm^{-1} . I believe it should be mm^{-2} . Please correct this error.

Thank you. Corrected.

3) In the methods section on page 13, there is a sentence saying, "Details on the worm simulation and video synthesis can be found in the methods section." The sentence refers to the method section even though it is within the methods section. Please revise this sentence for clarity.

Thank you. Corrected.

REVIEWERS' COMMENTS:

Reviewer #1 (Remarks to the Author):

The authors have addressed all of my concerns but one. When talking about head-tail assignment. The authors now state

"Notice that due to the flip-invariance of the distance metric, consecutive assignments of sets of centerlines are not necessarily aligned, hence a trivial alignment of the centerlines is carried out during post-processing before being analysed."

I understand that the alignment of consecutive centerlines to each other is trivial but the alignment to the actual worm head vs tail requires some more information. At this resolution I don't believe there will be any difference in appearance of the head and the tail and so determining which end of each tracked object is the head would require some other information (e.g. the head curvature is higher, or the amplitude is higher, or worms tend to swim forwards so the head is assumed to be the end that the centroid moves towards the most). Are these kinds of heuristics used in the final post-processing or some other method? Or is the head-tail assignment in fact random across worms?

Otherwise the paper remains excellent.

Reviewer #2 (Remarks to the Author):

The authors have thoughtfully addressed my queries and comments, and I am confident that this work will be an intriguing tool for the research community. It has the potential to benefit not only those studying *C. elegans* but also researchers tracking slender worm-like organisms and possibly filamentous objects.

-Ishant Tiwari